# Association of SARS-CoV-2 Seropositivity with Persistent Immune Activation in HIV/Tuberculosis Co-Infected Patients

**DOI:** 10.3390/reports7030061

**Published:** 2024-07-29

**Authors:** Ashwini Shete, Manisha Ghate, Hiroko Iwasaki-Hozumi, Sandip Patil, Pallavi Shidhaye, Takashi Matsuba, Gaowa Bai, Pratiksha Pharande, Toshio Hattori

**Affiliations:** 1Indian Council of Medical Research—National Institute of Translational Virology and AIDS Research (ICMR-NITVAR, Formerly National AIDS Research Institute), Pune 411026, India; mghate@nariindia.org (M.G.); spatil@nariindia.org (S.P.); pshidhaye@nariindia.org (P.S.); pratiksha.pharande1901@gmail.com (P.P.); 2Research Institute of Health and Welfare, Kibi International University, Takahashi 716-0018, Japan; hiro_ihz@kiui.ac.jp (H.I.-H.); bgw2291@imau.edu.cn (G.B.); 3School of Pharmaceutical Science, Kyushu University of Medical Sciences, Nobeoka 882-8508, Japan; matsubat@phoenix.ac.jp; 4College of Food Science and Engineering, Inner Mongolia Agricultural University, Hohhot 010018, China; 5The Shizuoka Graduate University of Public Health, Shizuoka City 420-0881, Japan

**Keywords:** SARS-CoV-2, HIV/tuberculosis co-infection, inflammatory markers, persistent immune activation, inflammatory score

## Abstract

We asked if SARS-CoV-2 seropositivity in HIV/TB co-infected patients plays a role in precipitating active tuberculosis in HIV-infected individuals and alters inflammatory status. A prospective study was conducted on HIV/TB co-infected patients presenting with pulmonary (*n* = 20) or extrapulmonary (*n* = 12) tuberculosis. Abbott SARS-CoV-2 IgG kits assessed the presence of anti-nucleoprotein antibodies. Inflammatory markers viz. osteopontin, total and full-length galectin-9, and C-reactive protein were tested at baseline and the end of antituberculosis treatment. The inflammatory score (INS) was assessed based on the percentage of reduction in the inflammatory markers’ levels at the end of the treatment. Anti-SARS-CoV-2 antibodies were detected in five male patients diagnosed with pulmonary (*n* = 2) and extrapulmonary (*n* = 3) TB. None of them reported symptomatic COVID-19. Inflammatory marker levels did not differ significantly at baseline compared to those in seronegative patients. However, the INS correlated negatively with SARS-CoV-2 seropositivity (*r* = −0.386, *p* = 0.039), indicating persistently raised inflammatory markers in these patients at the end of the treatment compared to seronegative individuals. Among the four markers studied, total galectin-9 levels failed to decrease significantly in these patients (*p* = 0.030). The majority of HIV/TB co-infected patients enrolled in our study (84.5%) were SARS-CoV-2-seronegative, indicating that SARS-CoV-2 infection might not have played a role in precipitating TB reactivation.

## 1. Introduction

Tuberculosis (TB) is a major public health threat recognized worldwide and, hence, had been prioritized during the development of the Millennium Development Goals (MDGs) by the United Nations (UN) [1]. Human immunodeficiency virus (HIV) infection is the most potent risk factor for developing TB and causing its resurgence in the African continent [2,3]. The course and management of both of these infections were severely affected during the pandemic caused by severe acute respiratory syndrome coronavirus 2 (SARS-CoV-2) [4]. In 2020, there were roughly 1.5 million tuberculosis deaths worldwide, representing the first year-over-year increase in TB deaths since 2005 [5]. Although TB notifications in India declined during the COVID-19 waves, a 19% increase was observed in 2021 from the previous year in TB patients’ notification, as per the India TB report 2022 [6]. 

The mechanisms of interaction between SARS-CoV-2 and HIV or TB infections in these clinical settings are not precise. It was proposed that the reactivation of latent tuberculosis in post-SARS-CoV-2-infected patients occurs because CD4+ T cells are exhausted and reduced in COVID-19 patients [7]. TB infections were reported to aggravate the symptoms of COVID-19 infections and prolong the viral shedding period of COVID-19 patients [8]. New Yorkers with HIV experienced an elevated risk for poor COVID-19 outcomes compared to those without HIV in 2020, indicating that people living with HIV (PLWH) must be attended to with priority, particularly those with low CD4 counts or underlying conditions [9]. A higher prevalence and incidence of TB among PLWH was also reported in Thailand during the COVID-19 pandemic [10]. In India, males, those with HIV–TB co-infection, the senior population, pulmonary TB patients, and those with MDR TB were at a higher risk of death from COVID-19 infection [11]. Furthermore, the substantial prevalence of post-COVID-19 conditions (PCCs) among PLWH was noted three months post-SARS-CoV-2 infection [12]. 

Matricelluar proteins play a very important role in the pathogenesis of various diseases [13,14] and have been implicated in mediating the severity of HIV and TB infections [15,16,17,18]. Importantly, they play a crucial role in the process of granuloma formation and progression in tuberculosis. Osteopontin (OPN) has been shown to serve as a reliable plasma marker to monitor TB activity [19]. It is expressed in granulomas and acts as a chemoattractant for various immune cells, causing local inflammation, tissue damage, and necrosis [20,21]. Its persistently raised levels indicate ongoing inflammation, possibly contributing to further tissue damage. Chronic increases in the levels of these proteins also cause systemic effects and have been implicated in mediating various other co-morbidities such as atherosclerosis and cardiovascular diseases [22,23]. It is known that OPN levels predict adverse outcomes in COVID-19 patients [24]. The level is also reported to be associated with post-acute-COVID-19-related dyspnea [25]. Similarly, Galectin-9 (Gal-9) has also been implicated in the severity of post-COVID-19 pulmonary fibrosis in SARS-CoV2- infections [18]. Hence, monitoring the levels of these proteins is of the utmost importance to ensure the recovery of the infected individuals and to predict post-recovery complications. We earlier reported a system to monitor immunologic recovery in HIV/TB co-infected patients [26] by assessing the degree of reduction in the levels of these proteins along with the other important inflammatory marker, C-reactive protein (CRP), which has also been implicated in predicting the severity of TB and COVID-19 [27,28]. Since even mild SARS-CoV-2 infections may lead to persistent inflammation resulting in long-COVID syndrome, we assessed the effect of possible SARS-CoV-2 infections on the inflammatory score indicative of immunological recovery in HIV/TB co-infected patients enrolled in our study [29]. The findings were partly presented at the international conference of the Indian Virological Society, VIROCON 2023 [30]. 

## 2. Materials and Methods

### 2.1. Study Settings and Participants

This prospective study was conducted in a cohort of HIV/TB co-infected individuals undergoing treatment at ART centers in Pune in India from January 2022 to December 2023. The participants were enrolled before or within 14 days of the initiation of antituberculosis treatment. We enrolled individuals with pulmonary (*n* = 20) or extrapulmonary (*n* = 12) tuberculosis. Pulmonary tuberculosis was confirmed microbiologically using the sputum acid-fast bacillus test (AFB) and/or the cartridge-based nucleic acid amplification test. However, it was not possible to confirm the diagnosis of extrapulmonary tuberculosis microbiologically and hence, it was primarily based on clinical findings and/or radiological investigations [31].

### 2.2. Sample Collection and Data Collection

The participants were followed up for collection of clinical data and blood samples at three time-points spanning a period of their antituberculosis therapy. The time-points included baseline (V1), at the end of the intensive phase of the treatment corresponding to the second month of the treatment (V2), and at the end of treatment (V3). Data of their routine radiological and laboratory investigations were noted from their treatment cards. 

### 2.3. Assessment of Matricellular Proteins, Other Inflammatory Markers, Cytokines, and Anti-SARS-CoV-2 Antibodies

The patients’ inflammatory conditions were monitored by measuring the levels of OPN (R&D Systems, Minneapolis, MN, USA), total Galectin-9 (T-Gal9) (R&D Systems, Minneapolis, MN, USA), intact full-length Galectin-9 (FL-Gal9) (ELISA Genie, Dublin, Ireland), and C-reactive protein (CRP) (Bio Check, San Francisco, CA, USA) in their plasma samples. The values were determined using commercially available ELISA kits at the above-mentioned three time-points, as described previously [16,26]. OPN and T-Gal9 were measured using the ELISA kits (R&D Systems, Minneapolis, MN, USA), which detect both full-length and cleaved forms, as described before [32,33]. Levels of cytokines such as IL-5, IL-4, GM-CSF, IL-10, IFN-γ, IL-21, TNF-α, IL-12 (p70), IL-2, and IL-17A were estimated using the multiplex luminex-based assay (Merck Millipore, Burlington, MA, USA) using the Bio-Plex 200 system (Bio-Rad, Hercules, CA, USA). Abbott SARS-CoV-2 IgG kits measured SARS-CoV-2 antibodies. 

### 2.4. Calculation of Inflammatory Score (INS)

We calculated the percentage decrease rate for each of the above-mentioned four markers. It was calculated by using the following formula:
(Concentration at the 1st visit minus concentration at the 3rd visit)/Concentration at the 1st visit × 100

The percent decrease rate for each marker was further scored as follows: >75%: 3; between 75 and 50%: 2; between 50 and 25%: 1; between 25 and −25%:0, ≤−25%: −1. The final INS was calculated by adding the individual scores for these markers.

### 2.5. Statistical Analysis

GraphPad Prism version 8 (San Diego, CA, USA) was used for performing statistical analysis. The differences between SARS-CoV-2-seropositive and -negative groups were assessed using the Mann–Whitney U test. Spearman’s rank correlation coefficient assessed correlations.

## 3. Results

### 3.1. Characterization of the Patients

We assessed samples of the HIV/TB co-infected participants (*n* = 32) enrolled in our study for the presence of anti-SARS-CoV-2 antibodies. Of the 32 individuals, 5 patients (2 with pulmonary TB and 3 with extrapulmonary TB) showed the presence of antibodies to SARS-CoV-2. There were no differences in clinical and laboratory findings between antibody-positive and -negative cases (Table 1).

### 3.2. Therapeutic Effects on Inflammatory Markers in Seronegative and Seropositive Groups

All four markers (T-Gal9, FL-Gal9, OPN, and CRP) significantly decreased in seronegative individuals (*p* = 0.0003, <0.0001, <0.0001, and 0.0004, respectively) at the end of the treatment as compared to their baseline levels. Among the seropositive groups, the decrease was not significant (Figure 1). 

### 3.3. Inverse Association of SARS-CoV-2 Seropositivity with Changes in the Levels of the Inflammatory Markers at the End of ATT

We have compared the percent change in four makers between seronegative and seropositive groups. The percent change was significantly different in T-Gal9 levels (*p* = 0.0319), with anti-SARS-CoV-2-seronegative individuals having a significantly greater decline than the seropositive individuals (Figure 2A). The percent changes were not significant for the other three markers. The INS was also considerably higher (*p* = 0.0412) in anti-SARS-CoV-2 seronegative individuals than in seropositive individuals (Figure 2B). 

We performed a correlation analysis of SARS-CoV-2 seropositivity with INS and T-Gal9 levels to confirm their associations (Table 2). Seropositivity was correlated negatively with INS and positively with T-Gal9 levels (*r* = −0.386, *p* = 0.039 and *r* = −0.404, *p* = 0.030, respectively).

### 3.4. Association of SARS-CoV-2 Seropositivity and INS with Changes in the Levels of Various Cytokines at the End of ATT

We further performed a correlation analysis of SARS-CoV-2 seropositivity and INS with the levels of various cytokines to assess the associations between them (Table 3). A significant correlation was observed between baseline IL-5 levels and SARS-CoV-2 seropositivity (*r* = −0.379, *p* = 0.015). The rate of decrease in IL-10 levels at the end of ATT was found to correlate negatively with the INS (*r* = −0.398, *p* = 0.030), while SARS-CoV-2 seropositivity was found to correlate positively with the rate of decrease in IL-4, IL-21, and IL-12 (p70).

## 4. Discussion

This study allowed us to observe the effects of a previous COVID-19 infection on the plasma inflammatory molecules of patients with HIV/TB. We investigated the presence of SARS-CoV-2 seropositivity in pulmonary and extrapulmonary TB cases enrolled in our study. Anti-SARS-CoV-2 nucleoprotein antibodies were detected in only five patients in our cohort with pulmonary (*n* = 2) or extrapulmonary (*n* = 3) TB. SARS-CoV-2 seroprevalences determined in India during the second wave of the pandemic to obtain information about the disease burden, as well as susceptible population, showed around 66% and 38% seroprevalence with the assessment of anti-spike and anti-nucleocapsid antibodies, respectively [34]. The higher seroprevalence of anti-spike antibodies was attributed to COVID-19 vaccination coverage. It has also been shown that anti-nucleocapsid antibodies persist for a shorter duration than anti-spike antibodies [35] and might indicate the presence of a recent infection. We wanted to rule out a recent SARS-CoV-2 infection, as the median time to develop TB after COVID-19 is reported to be 4 days although it may take up to seven months after COVID-19 recovery to develop tuberculosis, as reported in one of the published systemic reviews [36]. Anti-nucleocapsid antibody seropositivity has been reported to decrease from more than 90% after 3 months to more than 70% at 18 months, indicating their presence in most of the individuals up until 18 months [37]. Hence, missing SARS-CoV-2-infected cases due to the detection of anti-nucleocapsid antibodies were less likely. A slightly lower seropositivity of 54.6% in HIV-infected individuals attending ART centers affiliated with our institute than that reported in the general population was observed using a commercially available kit detecting both anti-spike as well as anti-nucleocapsid antibodies [38]. One of the studies has reported fewer SARS-CoV-2 infections in PLWH than those without HIV with lower IgG concentrations, possibly reflecting a diminished serological response to the infection [39]. Contrarily, a comparable seroprevalence in PLWH and the general population has also been reported in multiple studies with similar rates of seroconversion and stability [40,41]. However, the role of the diminished serological responses to SARS-CoV-2 due to HIV infection in the low seropositivity detected in our cohort cannot be ruled out. Although serological assays are the mainstay for determining the prevalence of past infections, there are studies reporting detectable T cell responses, especially in the settings of asymptomatic infections with low or undetectable antibodies, indicating the utility of the assessment of T cell responses for the detection of individuals with a previous infection [42]. However, we conducted this study in HIV-infected individuals where T cell responses may be suppressed due to immunodeficiency [43], and hence, we preferred serological assays for detecting the past infections. 

Long-COVID or post-COVID conditions (PCCs) are reported in around one-third of COVID-19 patients experiencing persistent symptoms lasting for more than three months after SARS-CoV-2 infection. In the HIV cohort study, PCC prevalence was 34.9%, occurring around three months post-SARS-CoV-2 infection, irrespective of their CD4 counts, viral loads, and immunization [12]. It was proposed that the reactivation of latent TB may occur as a part of the PCCs. Possible mechanisms that may contribute to such a reactivation have been thought to be related to immunosuppressive conditions caused by corticosteroids, higher pro-inflammatory responses, and the depletion of T cells [44,45]. Such a reactivation of TB was reported to occur in the lungs, as well as at extrapulmonary locations [36]. The majority of HIV/TB co-infected patients enrolled in our study (84.5%) were SARS-CoV-2-seronegative, indicating SARS-CoV-2-infection-mediated immunosuppression might not have played a role in precipitating TB reactivation in them. Our five patients did not show symptoms of COVID-19 and probably suffered from mild or subclinical SARS-CoV-2 infections. This could be one of the reasons why we did not observe significant differences in the levels of inflammatory markers in seropositive versus seronegative patients at baseline. Also, immunosuppressed PLHW were less likely to mount a cytokine storm and had lower inflammatory marker elevation during COVID-19 infection, as observed by one of the studies [46], indicating that raised pro-inflammatory cytokine levels might not play a role in TB reactivation in PLWH. A plausible mechanism contributing to reactivation is the depletion of CD4+ T cells in COVID-19 patients, which mediate the vital immune responses against mycobacteria [7]. Since our patients were infected with HIV, their CD4 levels were already low and did not differ from the seronegative group. HIV-negative TB patients were also reported to have lower Th1- and Th2-associated cytokines in serum, as well as in response to mycobacterial antigens [47,48], indicating that immunosuppression in the co-infected individuals might contribute to TB reactivation. Baseline levels of most of the cytokines in our study also correlated negatively with SARS-CoV-2 seropositivity. All patients with SARS-CoV-2 seropositivity in our study were males. In meta-analytical studies, male preponderance has been reported in COVID-19 and TB co-infections [49,50].

Our study demonstrated that inflammatory response was more likely to persist in those with SARS-CoV-2 seropositivity. Previous research has revealed that MCP proteins such as OPN and Gal-9 correlate with these diseases’ severity [16,17,24,51]. However, knowing which inflammatory protein is responsible for inflammatory damage is difficult, especially in multiple co-infections. Here, we have scored these complex reactions according to the INS. We selected four proteins known to be markers of these diseases—CRP, OPN, FL-Gal-9, and T-Gal9 [26]. Like T-Gal9, OPN was measured for both the truncated and FL forms, as these values often reflected the pathology of the diseases [15,33]. Interestingly, the difference between the two groups was observed in the INS. The INS remained low after therapy, indicating that an inflammatory state persisted in the seropositive group. Although the mechanism behind this difference is unknown, it suggests that cells producing MCP proteins may be subtly activated. A lower INS indicates persistent immune activation, which might contribute to the development of long-term co-morbidities in them. Further careful clinical observation of PLWH, along with monitoring the INS, would help understand their immune activation status. The role of MCPs in mediating PCCs has been demonstrated by one of the studies, which showed an association between plasma OPN levels and the persistence of severe exertional dyspnea and poor quality of life in COVID-19 patients who required hospitalization [25]. Furthermore, increased circulating Gal-9 levels along with increased monocyte activation were shown to be associated with the severity of fatigue in long-COVID patients [52]. We observed that T-Gal9 levels, but not FL-Gal9 levels, failed to decrease significantly in the seropositive patients. The cleaved Gal-9 represented in total Gal-9 has been shown to induce higher immune perturbations than those caused by full-length proteins [53]. T-Gal9 levels are shown to induce HIV viremia and also to contribute to inflammation [54]; hence, failure to reduce these levels post-treatment might be of a significant clinical concern. 

We further assessed levels of various cytokines at the baseline and at the end of ATT to determine their correlations with the INS, as well as with SARS-CoV-2 seropositivity. We found an inverse correlation of IL-5 levels with SARS-CoV-2 seropositivity, which was a surprising finding for us. Higher IL-5 levels were reported in COVD-19 patients infected with the wild-type virus previously [55]. However, IL-5 levels were much lower in patients infected with the delta variant of SARS-CoV-2, which caused the second wave of the pandemic in India [56]. IL-5 levels have been shown to correlate with Treg cells in HIV infection [57]. We observed a negative correlation of the levels with the INS, although they just missed the level of significance, indicating their role in mediating persistent immune activation. IL-10, an immunosuppressive cytokine secreted by Treg cells [58], was found to ameliorate the persistent immune activation in these patients, as evident by the negative correlation of the INS with the rate of decrease in its levels. We also report herewith the significant correlation of the rate of decrease in IL-4, IL-21, and IL-12 levels with SARS-CoV-2 seropositivity. The TH2 type of responses, which include cytokines such as IL-4/IL-21, are associated with the suppression of inflammatory responses [59]. Hence, it would be interesting to further investigate the role of higher decrease rates of these cytokines with persistent immune activation observed in our patients. 

The shortcomings of this paper are the inability to record the status of positive cases at the time of infection and a low number of seropositive cases, which caused a disparity in the number of individuals included in the seropositive and seronegative groups for analysis. However, individuals enrolled in the groups were matched in terms of certain basic demographic parameters like age/sex, as well as CD4 counts, indicating their comparability. Further levels of the immunological parameters studied by us are likely to be influenced by other co-morbidities to which HIV-infected individuals are highly prone However, we did not rule out all possible co-morbidities as they were not indicated based on the clinical manifestations. 

## 5. Conclusions

The proportion of HIV/TB co-infected participants enrolled in our study with SARS-CoV-2 seropositivity was below 20% and included both pulmonary and extrapulmonary tuberculosis. All these seropositive individuals were asymptomatic, and levels of the inflammatory markers assessed by us did not differ significantly at baseline between SARS-CoV-2-seropositive and -negative patients. However, the INS calculated based on the percent decrease correlated negatively with SARS-CoV-2 seropositivity, indicating persistently raised inflammatory markers in these patients at the end of the treatment compared to seronegative individuals. Also, among the four markers studied, T-Gal9 levels failed to decrease significantly in these patients, indicating the need for their close monitoring for assessing co-morbidities associated with persistent immune activation. 

## Figures and Tables

**Figure 1 reports-07-00061-f001:**
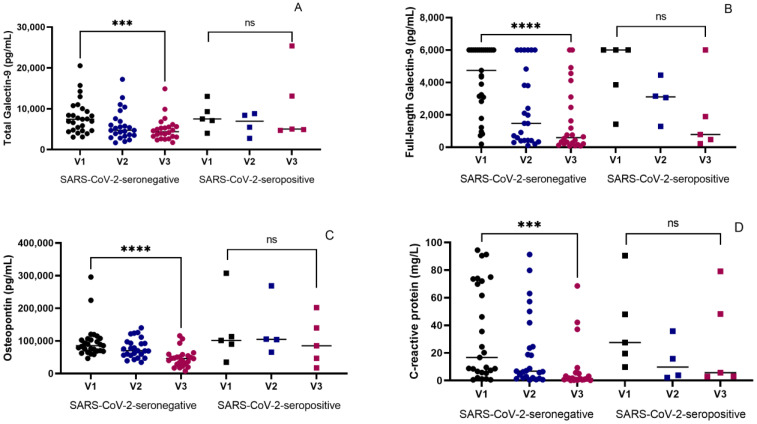
Changes in four inflammatory markers during three visits. V1, V2, and V3 are the 1st, 2nd, and 3rd visits. Dot plots showing changes in (**A**) plasma T-Gal9 levels, (**B**) plasma FL-Gal9 levels, (**C**) plasma OPN levels, and (**D**) plasma CRP levels in seronegative and seropositive groups at V1 (baseline), V2 (month 2), and V3 (end of treatment). *p* values showing significant changes in the levels are indicated as *** (*p* < 0.001) and **** (*p* < 0.0001). ns; not significant.

**Figure 2 reports-07-00061-f002:**
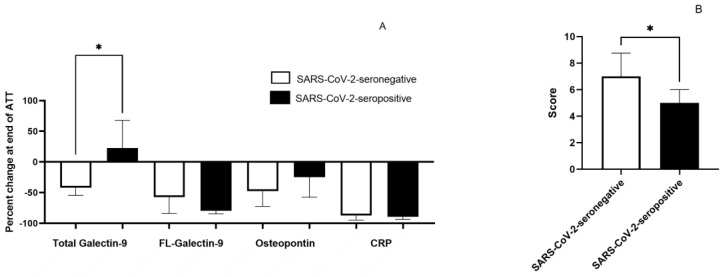
Changes in inflammatory markers after therapy. (**A**) Percent changes in the levels of the markers at the end of antituberculosis treatment in seropositive and seronegative groups. (**B**) The INS of seronegative and seropositive groups. * (*p* < 0.05).

**Table 1 reports-07-00061-t001:** Characteristics of the HIV/TB co-infected study participants concerning their SARS-CoV-2 serostatus at baseline.

Parameters, Median (IQR)	SARS-CoV-2-Seronegative Patients (*n* = 27)	SARS-CoV-2-Seropositive Patients (*n* = 5)	*p* Value
Age (Years)	45 (38–50)	48 (45–50)	NS
Gender—M:F	18:09	05:00	NS
Type—Pulmonary: extrapulmonary	18:09	2:3	NS
Body weight (Kg)	49 (39–56)	43.9 (40–66.5)	NS
Duration of therapy (months)	6 (6–11.75)	10 (6–12.5)	NS
CD4 count (cells/mL)	266 (141–387)	359 (261–509)	NS
T-Gal9 (pg/mL)	7439 (4657–10,038)	7507 (5561–11,172)	NS
FL-Gal9 (pg/mL)	4742 (2830–6000)	6000 (2638–6000)	NS
OPN (ng/mL)	85.5 (69.0–106.9)	101.5 (62.7–210.3)	NS
CRP (mg/L)	16.72 (5.77–72.05)	27.57 (14.69–69.23)	NS

**Table 2 reports-07-00061-t002:** Correlation of SARS-CoV-2 seropositivity with different markers.

Correlation Analysis	Correlation Coefficient (*r*)	*p* Value
SARS-CoV-2 seropositivity versus Inflammatory Score	−0.386	0.039
SARS-CoV-2 seropositivity versus percent change in total Galectin-9 levels	0.404	0.030

**Table 3 reports-07-00061-t003:** Correlation of baseline cytokine levels and their percent decrease with INS and SARS-CoV-2 seropositivity.

	Baseline Value				% Decrease Rate			
	Correlation with INS		Correlation with SARS-CoV-2 Seropositivity		Correlation with INS		Correlation with SARS-CoV-2 Seropositivity	
Cytokines	*r* Value	*p* Value	*r* Value	*p* Value	*r* Value	*p* Value	*r* Value	*p* Value
IL-5	−0.273	0.076	**−0.379**	**0.015**	0.041	0.421	0.319	0.056
IL-4	−0.128	0.258	−0.082	0.330	0.062	0.384	**0.364**	**0.037**
GM-CSF	−0.265	0.082	−0.192	0.142	0.005	0.491	−0.007	0.487
IL-10	0.080	0.343	0.005	0.490	**−0.398**	**0.030**	0.032	0.442
IFN-γ	−0.030	0.439	−0.238	0.095	0.054	0.400	0.097	0.322
IL-21	−0.308	0.055	−0.177	0.171	0.083	0.346	**0.406**	**0.022**
TNF-α	0.203	0.146	0.079	0.333	−0.022	0.459	−0.125	0.276
IL-12 (p70)	−0.175	0.182	−0.112	0.270	−0.189	0.177	**0.404**	**0.020**
IL-2	0.142	0.231	−0.219	0.111	0.169	0.210	0.325	0.057
IL-17A	−0.254	0.092	−0.037	0.420	0.049	0.408	0.250	0.114

Footnote—*p* values mentioned in the table are one-tailed, Bold values indicate significant *p* values.

## Data Availability

The original data presented in this study are available on reasonable request from the corresponding author. The data are not publicly available due to privacy.

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
