# Peer review of "Association of SARS-CoV-2 Seropositivity with Persistent Immune Activation in HIV/Tuberculosis Co-Infected Patients"

_reports, 2024, doi:10.3390/reports7030061_

Round 1
Reviewer 1 Report
Comments and Suggestions for Authors
The work by Dr. Shete explores the role of a previous SRAS-CoV-2 infection in precipitating tuberculosis (TB) infection in the context of HIV co-infection. This is a poorly explored issue; therefore, this study is highly relevant to the field.
The results showed by the authors points out that persistently raised inflammatory markers and regulatory/Th2 cytokines in SARS-CoV-2 seropositive patients at the end of anti-tuberculosis treatment correlated with SARS CoV-2 seropositivity, suggesting an increased inflammatory footprint accompanied by a counteracting anti-inflammatory response. It would have been interesting to see the impact of this immune landscape on the clinical resolution of TB infection, as measured by time of sputum-smear and/or culture negativity or another variable.
This reviewer thinks that the paper is relevant and should be published, but some improvements can be done.
- Point 2.3 (line 103). “Estimation” is not a proper term to describe the results depicted in that paragraph. Please use “Determination”, “Assessment” or another expression to describe the results from the quantification of soluble factors.
- Comparisons between seronegative and seropositive groups show disparity in the number of individuals in each group. This reviewer understands that it’s very difficult to reach equal numbers in both groups, but it would be interesting to see a statistical comparison between 5 (FIVE) perfectly matched (i.e. age, sex and CD4 count) individuals per group to confirm statistical significances.
- Discussion must be improved. Please discuss:
o The utility of cellular immunity detection (i.e., ELISPOT or flow cytometry) to search for patients with previous SARS-CoV-2 infection.
o The limitations of anti-N antibody detection in terms of duration of antibodies in plasma after SARS-CoV-2 infection. And the impact derived from this limitation on the results and conclusions of the work.
o In this regard, this reviewer considers extremely important to analyze and discuss the limitations of antibody detection in HIV-positive persons, due to the diminished serological responses to SARS-CoV-2.
o Another issue to be discussed (could be from the bibliography) is the level of inflammatory molecules and/or cytokines in HIV-positive patients without TB after SARS-CoV-2 infection? Could these levels be compared with those seen in HIV-TB patients after TB treatment completion? This analysis could shed some light on the question of what the impact of a previous SARS-CoV-2 infection on the development of TB within an HIV-infection background is.
- Line 251: misspelling “seropoitivity”. Please correct.
Comments on the Quality of English LanguageIt is a well-written manuscript. Some misspellings are detected.
Author Response
Reviewer 1
Comment 1
The work by Dr. Shete explores the role of a previous SRAS-CoV-2 infection in precipitating tuberculosis (TB) infection in the context of HIV co-infection. This is a poorly explored issue; therefore, this study is highly relevant to the field.
The results showed by the authors points out that persistently raised inflammatory markers and regulatory/Th2 cytokines in SARS-CoV-2 seropositive patients at the end of anti-tuberculosis treatment correlated with SARS CoV-2 seropositivity, suggesting an increased inflammatory footprint accompanied by a counteracting anti-inflammatory response. It would have been interesting to see the impact of this immune landscape on the clinical resolution of TB infection, as measured by time of sputum-smear and/or culture negativity or another variable.
This reviewer thinks that the paper is relevant and should be published, but some improvements can be done.
- Point 2.3 (line 103). “Estimation” is not a proper term to describe the results depicted in that paragraph. Please use “Determination”, “Assessment” or another expression to describe the results from the quantification of soluble factors.
Response 1
Thank you for your kind editing. It was corrected.
Comment 2
- Comparisons between seronegative and seropositive groups show disparity in the number of individuals in each group. This reviewer understands that it’s very difficult to reach equal numbers in both groups, but it would be interesting to see a statistical comparison between 5 (FIVE) perfectly matched (i.e. age, sex and CD4 count) individuals per group to confirm statistical significances.
Response 2
We understand the limitation of the study owing to the disparity in the number of individuals included in each group, which we have explicitly mentioned as a limitation in the discussion section (Page 8). We have included a comparison of these two groups in terms of parameters such as age, sex, and CD4 counts. We could not find any statistical significance indicating matching of these two groups. The data is presented in Table 1 of the manuscript.
Comment 3
- Discussion must be improved. Please discuss:
- The utility of cellular immunity detection (i.e., ELISPOT or flow cytometry) to search for patients with previous SARS-CoV-2 infection.
Response 3
We would like to thank the reviewer for the suggestion. Serological assays are the mainstay for determining prevalence of past infections. However, there are studies reporting detectable T cell responses, especially in the settings of asymptomatic infections with low or undetectable antibodies, indicating the utility of assessment of T cell responses for the detection of individuals with previous infection. We conducted this study in HIV-infected individuals where T cell responses may be suppressed due to immunodeficiency, leading to the possibility of missing past infections.
Discussion related to it has now been added to the manuscript (Page 7).
Comment 4
- The limitations of anti-N antibody detection in terms of duration of antibodies in plasma after SARS-CoV-2 infection. And the impact derived from this limitation on the results and conclusions of the work.
Response 4
Compared to anti-S antibody responses, those against the N protein appear to wane in the post infection cohort. We wanted to rule out recent SARS Cov2 infection as the median time to develop TB after COVID-19 is 4 days, although it may take up to seven months after COVID-19 recovery to develop tuberculosis as reported in one of the published systemic reviews (Alemu, A., et al., 2022). Seropositivity has reported to decrease from more than 90% after 3 months to more than 70% at 18 months. Hence, missing SARS Cov2 infected cases due to detection of anti-N antibody is a less likely possibility.
Discussion related to it has now been added to the manuscript (Page 6).
Comment 5
- In this regard, this reviewer considers extremely important to analyze and discuss the limitations of antibody detection in HIV-positive persons, due to the diminished serological responses to SARS-CoV-2.
Response 5
We would like to thank the reviewer for bringing up this important limitation. A slightly lower seropositivity of 54.6% in HIV-infected individuals attending ART centers affiliated with our institute than that reported in the general population was observed using a commercially available kit detecting both anti-spike as well as anti nucleocapsid antibodies. One of the studies has reported fewer SARS-CoV-2 infections in PLWH than those without HIV with lower IgG concentrations possibly reflecting a diminished serological response to infection. Contrarily, a comparable seroprevalence in PLWH and general population has also been reported in multiple studies with similar rates of seroconversion and stability. However, role of the diminished serological responses to SARS-CoV-2 due to HIV infection in the low seropositivity detected in our cohort cannot be ruled out.
Discussion related to it has now been added in the manuscript (pages 6-7).
Comment 6
- Another issue to be discussed (could be from the bibliography) is the level of inflammatory molecules and/or cytokines in HIV-positive patients without TB after SARS-CoV-2 infection? Could these levels be compared with those seen in HIV-TB patients after TB treatment completion? This analysis could shed some light on the question of what the impact of a previous SARS-CoV-2 infection on the development of TB within an HIV-infection background is.
Response 6
There are no reports measuring OPN or Gal-9 in HIV/COVID-19 and ours is the first study reporting levels of these matricellular proteins in the triple infection. Immunosuppressed PLHW were reported to less likely mount a cytokine storm and had lower inflammatory marker elevation during COVID-19 infection indicating raised proinflammatory cytokine levels might not play a role in TB reactivation in PLWH.
HIV neg TB patients also were reported to have lower Th1- and Th2-associated cytokines in serum as well as in response to mycobacterial antigens indicating immunosuppression in the coinfected individuals might have contributed to TB reactivation.
Discussion related to it has now been added in the manuscript (page 7).
Comment 7
Line 251: misspelling “seropoitivity”. Please correct.
Response 7
Misspelling is corrected now.
Comment 8
Comments on the Quality of English Language
It is a well-written manuscript. Some misspellings are detected.
Response 8
Misspellings are corrected now.

Reviewer 2 Report
Comments and Suggestions for Authors
The authors investigated the effects of previous SARS-CoV-2 virus infection on inflammatory molecules in HIV/TB coinfected patients. The authors concluded that the SARS-CoV-2 virus infection might not have played a role in precipitating TB reactivation.
Please see the main concerns below.
1. For markers (T-Gal9, FL-Gal9, OPN, and CRP) were selected in this study. what is the rational of selecting these four markers?
2. Anti- SARS-CoV-2 nucleoprotein antibodies were detected in five patients. It seems this number is low and may not provide statistics significance.
3. The anti-nucleoprotein antibodies were detected in five patients, how long will these antibodies last? Is it possible that patients were infected by SARS-CoV-2 virus, but antibodies are undetectable when experiments performed?
Author Response
Reviewer 2
Comment 1
The authors investigated the effects of previous SARS-CoV-2 virus infection on inflammatory molecules in HIV/TB coinfected patients. The authors concluded that the SARS-CoV-2 virus infection might not have played a role in precipitating TB reactivation.
Please see the main concerns below.
- For markers (T-Gal9, FL-Gal9, OPN, and CRP) were selected in this study. what is the rational of selecting these four markers?
Response 1
We are sorry for not explicitly clarifying the selected markers' role in our manuscript.
Matricelluar proteins play a very important role in the pathogenesis of various diseases and have been implicated in mediating the severity of HIV and TB infections. Importantly, they play a very important role in the process of granuloma formation and progression in tuberculosis. OPN has been shown to serve as a reliable plasma marker to monitor TB activity. They are expressed in on granulomas and act as chemoattractants for various immune cells causing local inflammation, tissue damage and necrosis. Their persistently raised levels indicate ongoing inflammation and might contribute to further tissue damage. Chronic increases in the levels of these proteins also cause systemic effects and have been implicated in mediating various other co-morbidities such as atherosclerosis, and cardiovascular diseases. It is known that OPN levels predict adverse outcomes in COVID-19 patients. The level is also reported to be associated with post-acute COVID-19-related dyspnea. Similarly Galectin-9 has also been implicated in the severity and post-COVID-19 pulmonary fibrosis in SARS-CoV2- infections. Hence monitoring levels of these proteins is of utmost importance to ensure the recovery of the infected individuals and predict post-recovery complications. We earlier reported a system to monitor immunologic recovery in HIV/TB coinfected patients by assessing the degree of reduction in the levels of these proteins along with the other important inflammatory marker, C-reactive protein which also has been implicated in predicting the severity of TB and COVID-19.
A description related to it has now been added in the introduction section with the relevant references (page 2).
Comment 2
- Anti- SARS-CoV-2 nucleoprotein antibodies were detected in five patients. It seems this number is low and may not provide statistics significance.
Response 2
We accept that a low number of seropositive cases is one of our study’s limitations and have mentioned it in the discussion part (Page 8). We feel that the findings are still important as the study was conducted prospectively, and there are very few reports on HIV/TB/SARS-CoV-2 triple infections. We have also analyzed the data as a case-control design to get meaningful data with a smaller sample size.
Comment 3
- The anti-nucleoprotein antibodies were detected in five patients, how long will these antibodies last? Is it possible that patients were infected by SARS-CoV-2 virus, but antibodies are undetectable when experiments performed?
Response 3
It has also been shown that anti-nucleocapsid antibodies persist for a shorter duration as compared to anti-spike antibodies and might indicate the presence of a recent infection. We wanted to rule out recent SARS Cov2 infection as median time to develop TB after COVID-19 is reported to be 4 days although it may take up to seven months after COVID-19 recovery to develop tuberculosis as reported in one of the published systemic reviews. Anti-nucleocapsid antibodies seropositivity has been reported to decrease from more than 90% after 3 months to more than 70% at 18 months indicating their presence in most of the individuals till 18 months. Hence missing SARS Cov2 infected cases due to detection of anti-N antibody was a less likely possibility.
Discussion related to it has now been added to the manuscript (Page 6).

Reviewer 3 Report
Comments and Suggestions for Authors
The manuscript is very difficult to read. Both the introduction and the discussion contain a lot of information that is not directly related to the aim of the work. There is no clear presentation of a specific main idea – what authors are trying to study and what to compare with. The introduction lists a few ideas. In such a small-scale study, in my opinion, there should be one clear, well-founded idea.
Now there is no such single idea, so it is difficult to assess the course of the study itself and the criteria for patient selection.
The most important markers of the study, osteopontin, and galectin-9, are highly non-specific. They are expressed in many diseases. Therefore, to investigate the value of these markers, the study population should be highly homogeneous. In this case, nothing is known about the co-morbidities of the patients. Patients with both tuberculosis and HIV usually have at least a few various co-morbidities.
The diagnosis of tuberculosis is inappropriate. The standard criterion for the diagnosis of tuberculosis is tuberculosis mycobacteria culture. In some cases – a rapid molecular method such as Xpert MTB/RIF. The acid-fast bacillus test is not a diagnostic method, as it does not allow to distinguish tuberculosis mycobacteria from non-tuberculous mycobacteria and some other microorganisms.
Author Response
Reviewer 3
Comment 1
The manuscript is very difficult to read. Both the introduction and the discussion contain a lot of information that is not directly related to the aim of the work. There is no clear presentation of a specific main idea – what authors are trying to study and what to compare with. The introduction lists a few ideas. In such a small-scale study, in my opinion, there should be one clear, well-founded idea.
Now there is no such single idea, so it is difficult to assess the course of the study itself and the criteria for patient selection.
Response 1
As a case report journal, Reports covers a wide variety of diseases, and many of its readers are not infectious disease specialists. As a case report journal, we included public health facts to explain the importance of the disease. However, as you pointed out, this could have been a distraction from the main point of the paper, so we shortened the public health description a little. Our argument is not many, but one: that is to assess the effect of possible SARS-CoV-2 infections on the inflammatory score indicative of immunological recovery in HIV/TB coinfected patients.
However, because this argument was written at the very end of the discussion, it is likely to cause confusion. We decided to put this claim in writing as early as possible during the discussion. We have modified and reorganized the introduction and discussion to bring out our argument clearly.
Comment 2
The most important markers of the study, osteopontin, and galectin-9, are highly non-specific. They are expressed in many diseases. Therefore, to investigate the value of these markers, the study population should be highly homogeneous. In this case, nothing is known about the co-morbidities of the patients. Patients with both tuberculosis and HIV usually have at least a few various co-morbidities.
Response 2
We agree that the immunological biomarkers are highly non-specific and are raised in multiple diseases. Studies in human population is complex as there are multiple confounding factors which can affect the results. Hence we have analyzed the findings as a case control design in the two groups where SARS-CoV-2 seropositive individuals served as cases while SARS-CoV-2 negative individuals acted as controls. There was no selection bias as the SARS-CoV-2 antibody testing was conducted at a later date. Both the groups were comparable in their basic characteristics as shown in the table 1. We clinically examined them for the presence of other opportunistic infections and other co-morbidities. However, we did not perform tests to rule out other conditions which were not clinically indicated.
HIV infected individuals can have multiple co-morbidities and it is not feasible to rule out each of them due to various constraints. Hence we investigated as per the recommendations of our national program. But we understand this as one of the limitations and hence we have described it in the discussion part (page 8).
Comment 3
The diagnosis of tuberculosis is inappropriate. The standard criterion for the diagnosis of tuberculosis is tuberculosis mycobacteria culture. In some cases – a rapid molecular method such as Xpert MTB/RIF. The acid-fast bacillus test is not a diagnostic method, as it does not allow to distinguish tuberculosis mycobacteria from non-tuberculous mycobacteria and some other microorganisms.
Response 3
We agree with the reviewer’s comments regarding the diagnosis of tuberculosis. Although the acid-fast bacillus test does not distinguish between tuberculosis mycobacteria from non-tuberculous mycobacteria and some other microorganisms, it is still the most widely used method for detecting TB. We use primarily rapid molecular method such as Xpert MTB/RIF for diagnosing TB in HIV infected patients. However, Sputum Smear Microscopy for AFB detection is also acceptable for microbiological diagnosis as per our National Tuberculosis Elimination program (https://www.ntep.in/node/1743/CP-diagnostic-algorithm-tb-disease-ntep)

Round 2
Reviewer 2 Report
Comments and Suggestions for Authors
this is an updated version and the authors have addressed the concerns
Reviewer 3 Report
Comments and Suggestions for Authors
None